# Characterization of Antibiotic Resistant Coliform Bacteria and Resistance Genes Isolated from Samples of Smoothie Drinks and Raw Milk

**DOI:** 10.3390/foods11091324

**Published:** 2022-05-01

**Authors:** Monika Krahulcová, Klára Cverenkárová, Petra Olejníková, Barbora Micajová, Júlia Koreneková, Lucia Bírošová

**Affiliations:** 1Department of Nutrition and Food Quality Assessment, Faculty of Chemical and Food Technology, Slovak University of Technology, Radlinského 9, 81237 Bratislava, Slovakia; klara.cverenkarova@stuba.sk (K.C.); baskam@ymail.com (B.M.); xkorenekova@stuba.sk (J.K.); lucyja.birosova@gmail.com (L.B.); 2Institute of Biochemistry and Microbiology, Faculty of Chemical and Food Technology, Slovak University of Technology, Radlinského 9, 81237 Bratislava, Slovakia; petra.olejnikova@stuba.sk

**Keywords:** coliform bacteria, smoothie, raw milk, antibiotic resistance, resistance genes

## Abstract

Raw foodstuffs have been marked as a healthier alternative in the context of nutrient content and are becoming more popular with consumers. Thermally untreated foods may represent a microbiological risk connected with the possible presence of antimicrobial resistance. The aim of this study was to prove that popular raw food beverages such as smoothies and raw milk may be a source of antibiotic-resistant coliform bacteria and resistant genes. The majority of antibiotic-resistant isolates (110) were identified as *Enterobacter* spp., *Escherichia coli*, and species of *Klebsiella* spp., predominantly β-lactam and chloramphenicol resistant. Multidrug resistance has been registered in one-third of resistants. Overproduction of efflux pumps was clarified in 8 different bacteria. The majority of resistant isolates were strong biofilm producers. Antibiotic resistance gene *bla*_OXA_ was detected in 25% of isolates, especially in *E. coli*. Resistance genes *bla*_TEM_ and *bla*_SHV_ were detected in 19% and 14%, respectively. This is the first study to point out that popular raw drinks such as smoothies or raw milk, besides their nutrient benefits, could represent a reservoir of antibiotic-resistant bacteria as well as antibiotic resistance genes. According to this, raw drinks could contribute to the dissemination of antibiotic resistance in the human gastrointestinal tract and environment.

## 1. Introduction

Nowadays, one in five deaths is caused by an unhealthy diet [1]. Therefore, consumers around the world are becoming more and more aware of the benefits arising from a healthy diet. Modern lifestyles have seen new trends in eating habits, and types of diets are expanding [2]. Raw food consumption represents such a diet, as evidence suggests it offers more nutrients and enzymes, which could be destroyed by further processing [3]. On the other side, consumers should learn about the nutritional value of food and the potential of microbial contamination [4,5]. Traditionally, foodborne outbreaks were mainly connected to the consumption of food of animal origin (AO). Lately, foodborne epidemics caused by the consumption of plant-based food have repeatedly been appearing over time, which suggests that plant origin (PO) food is as dangerous as animal-based [5]. In 2020, plants such as leafy greens or clover sprouts reported 91 cases of foodborne illness caused by the bacterium *E. coli* O157:H7 [6,7]. In 2021, the bacterium was also detected in packaged salad and baby spinach [8,9]. During the period 2018–2019, the consumption of Romaine lettuce was the source of foodborne diseases leading to five deaths due to the presence of *E. coli* [10,11,12]. Smoothie drinks are prepared with such ingredients, and there is no further processing, which indicates a possible presence of such bacteria [13]. Unpasteurized milk represents another food commodity associated with the risk of ingesting pathogenic bacteria. Consumption of raw milk is not recommended, but there are some countries where the drinking of raw milk is still available in consideration of their traditions [14]. In a large-scale study, dairy was the second most common cause of foodborne outbreaks, where a majority of infection sources were identified as raw milk [15].

The presence of bacteria is directly connected to the occurrence of antibiotic resistance. Antibiotic resistance represents a serious challenge for modern medicine. According to the European Union, the rate of human deaths related to antibiotic-resistant bacterial infection was estimated at 33,000 per year. The majority of these infections were due to gram-negative bacteria [16]. World Health Organization (WHO) declaim that the number of deaths caused by drug-resistant bacterial strains could grow to 10 million per year by 2050, as more and more common diseases would be unthreadable due to multidrug-resistant (MDR) bacteria not responding to commercial clinical therapy [17]. One of the important mechanisms responsible for MDR is efflux pump overproduction. It reduces the number of antimicrobials by excluding the substance from the bacterial cell prior to the substance reaching the target in the bacterium [18,19]. Efflux pump overproduction may act simultaneously against different types of antibiotic (ATB) classes [19]. Antibiotic resistance is mainly spread through mobile genetic elements, within or between bacterial species, and from nonpathogenic to pathogenic strains [18]. Horizontal gene transfer (HGT) has a great impact and is considered as the most endangering way of spreading resistance. HGT may be mediated through microorganisms contained in manure, soil, or water, which are widely using for the growing of crops or vegetables, to feed animals, or through direct consumption of grown crops and vegetables by humans. Ingestion of such microorganisms with food may lead to a colonized digestive tract or infected humans [20].

An important property of microorganisms connected to food production is their ability to form biofilm. The production of biofilm may cause serious problems as it could result in food spoilage or, worse, become pathogenic [21]. Antimicrobial resistance, biofilm, represents an important feature in protecting bacterial cells. The formation of biofilm is a serious problem in hospital environments and causes the persistence of nosocomial infection. Species enclosed in biofilm are more resistant than planktonic ones [22]. Among the potentially pathogenic bacteria encountered in the environment are coliform bacteria, particularly *E. coli*. *Escherichia coli* is a common cause of nosocomial and community-acquired infections [23] and can pose various virulent factors [24,25]. Coliform bacteria present in some foods are considered indicators of poor hygiene, especially in foods that are not treated and protected by the effect of heating. The safety of raw food depends mainly on the hygiene standard of manipulation [26]. Coliform bacteria have protentional pathogenic nature if the conditions are appropriate. Additionally, representants such as *E. coli*, *Klebsiella* spp., or *Enterobacter* spp. have been repeatedly marked as extended-spectrum β-lactamases (ESBL) producing *Enterobacterales* [27]. ESBL are enzymes produced for protection against broad-spectrum β-lactam ATBs widely used in medical treatment. ESBL is one of the main reasons for the malfunction of ATB care as a new global upward trend [28].

Foodstuffs play an important role in the transmission of bacteria carrying resistance phenotypes [29,30]. There is evidence of similar or clonal-related antibiotic-resistant bacteria and resistance genes in humans, which could be connected to transfer through the direct consumption and/or indirect manipulation of food [18,20]. It is necessary to start to think in a “One health approach” to secure multi-sectoral integration, as all spheres of humans’ actions contribute to the problem of antibiotic resistance [31]. In recent studies, Krahulcová et al. (2018) and Krahulcová et al. (2021) have shown that raw ready-to-drink food such as smoothies or raw milk, which are popular in Slovakia, could be a source of antibiotic-resistant coliform bacteria [13,14]. Although many studies have pointed out the microbiological risk of such foodstuffs, our study emphasizes the presence of antibiotic-resistant bacteria in these popular drinks of PO and AO. In this respect, such food can contribute to the transfer of antibiotic-resistance genes in the human gut and, consequently, the dissemination of antibiotic resistance.

## 2. Materials and Methods

### 2.1. Sampling

Samples of raw milk were collected in sterile tubes from three milk vending machines located in the capital city of Slovakia, Bratislava, and one in the middle of Slovakia because this region has a tradition in the dairy industry. Collection was done in two periods: the winter (February–March) and summer seasons (August–September) of 2017. Microbial culturing was done immediately after transporting into the laboratory within 2–3 h from sampling. In addition, each sample was refrigerated (4–8 °C) during transport in a cooling box [14].

Six food-service establishments were chosen to monitor twenty samples of smoothies in Bratislava. Smoothie drinks were freshly prepared in the food-service establishments and were intended to be consumed in a short time (24 h). Such drinks were immediately transferred into the laboratory for microbiological analysis. The selection of different types of smoothies was performed according to consumer preferences to cover as many types as possible. Thirteen fruit-based smoothies and seven green-based smoothies were further analyzed. The common ingredients in green-based smoothies were spinach, ice lettuce, broccoli, or celery and in fruit-based smoothies were strawberry, banana, apple, or orange. Specifically, smoothies are summarized in studies performed by Krahulcová et al. (2021) [13].

### 2.2. Identification of Antibiotic-Resistant Strains

Antibiotic-resistant strains of coliform bacteria were identified by matrix-assisted laser desorption ionization-time of flight (MALDI-TOF) mass spectrometry (Bruker, Germany). Coliform bacteria showing antibiotic resistance were randomly selected and isolated by the streak plate method on Mueller Hinton agar (Biolife, Italy) plates for 24 h at 37 °C. The pure bacterial colony of each isolate was spotted on a steel target plate and dripped by a 1 μL matrix of α-cyano-4-hydroxycinnamic acid and left to air-dry. The matrix was prepared as a saturated solution in 2.5% trifluoroacetic acid and 50% acetonitrile. The target plate was inserted into MALDI-TOF mass spectrometry, and analyzation was performed via an AutoFlex I TOF-TOF apparatus (Bruker Daltonics Inc., Billerica, MA, USA) in linear positive-ion mode (*m*/*z* range of 2000 to 20,000 with gating of ions below *m*/*z* 400 and a delayed extraction time of 450 ns). Gaining spectra were analyzed using MALDI BioTyper software (v 2.0) based on an algorithm for matching spectral patterns in logarithmic scores 0–3 (BioTyper Library v 3.0; Bruker Daltonics s.r.o., Brno, Czech Republic). A score above 1.9 ensured bacterial species identification by comparison of the obtained bacterial fingerprints with the existing database [32].

### 2.3. Susceptibility Testing

The macro-dilution drop method was applied with resistant isolates to detect the susceptibility profile. The ATBs used for testing were ampicillin, ceftazidime, ciprofloxacin, tetracycline, gentamicin, chloramphenicol, and meropenem. The concentration of each ATB is listed in Table 1. Antibiotic concentration was defined by resistant breakpoints according to European guidelines established by The European Committee on Antimicrobial Susceptibility Testing EUCAST, marked as first concentration (R1) and according to American guidelines established by Clinical Laboratory Standards Institute CLSI, marked as second concentration (R2). The third concentration was selected to determine the highest level of resistance in isolates as it was chosen to exceed American standards. The experiment was performed using Mueller-Hinton agar (Biolife, Milan, Italy). Incubation of plates was at 37 °C for 24 h, and the evaluation of bacterial growth was visual [32]. Each experiment ran in triplicates and was repeated three times. For statistical analysis Student’s *t*-test was applied.

### 2.4. Biofilm Production Testing

According to Beenken et al. (2003) [33], exanimation of the ability to form biofilm was performed. Overnight cultures of resistant coliform isolates were diluted in a ratio of 1:200 to tryptic soy broth and inoculated to a sterile microtiter plate for static incubation for 24 h and 37 °C. After removing the overnight cultures, the wells were washed twice with 200 μL of phosphate-buffered saline (PBS). The forming biofilm was fixed with 96% ethanol in a volume of 200 μL of each well. The ethanol was immediately removed, and the microplate plate was dried at room temperature for about 10 min. Subsequently, biofilms at the bottom of wells were stained with crystal violet (0.41% in 12% ethanol). After 3 min of staining action, wells were again washed twice with PBS (200 μL of each). It the end, 96% ethanol was added to each well. The ability to form biofilm was measured using a plate reader device (BioTek Inc., Seattle, WA, USA) to gain absorbances at 570 nm. Each experiment was repeated three times and ran in six parallels. For statistical analysis Student’s *t*-test was applied. The positive control strain was *Pseudomonas aeruginosa* (CCM 3955), originating from the Czech Collection of Bacterial Strains in Brno, considered a strong producer of biofilm. By comparing the intensity of staining using the measured absorbances, the biofilm producers were evaluated on a scale of weak (<0.2), medium (0.2–0.3), strong (0.3–0.9), and very strong (>1.0) biofilm producers according to Taniguchi et al. (2009) [32].

### 2.5. Efflux Pumps Overproduction Testing with Ethidium Bromide (EtBr)

Detection of efflux pump overproduction was evaluated via the EtBr-agar Carthweel method [19]. First, each plate was divided into twelve equally sized sections according to the cartwheel pattern and marked properly. Agar plates were prepared with Mueller-Hinton (BioLife, Italy) agar containing ethidium bromide (EtBr) at a concentration of 2.5 mg/L to detect the overproduction of efflux pumps in gram-negative bacterial species. The plates should be protected from light and prepared the previous day. Overnight cultures of tested isolates were modified to the concentration of 0.5 McFarland standard and swabbed on EtBr-plates from the center to the edge of agar plates. Incubation of inoculated EtBr-plates was at 37 °C for 16 h. The experiment was based on visual evaluation with UV irradiation due to fluorescence active compound EtBr (Serva, Heidelberg, Germany) [34]. The reference strain used as a comparative negative control was *E. coli* (CCM 3988) from the Czech Collection of Bacterial Strains in Brno. Each experiment ran in triplicates and was repeated three times. For statistical analysis Student′s *t*-test was applied.

### 2.6. Antibiotic Resistance Genes Detection

Resistance genes were detected via single and multiplex polymerase-chain-reaction (PCR). Genes determined in the study were β-lactamases TEM, SHV, OXA, and CTX-M-group 1 [35] and tetracycline resistance genes Group II: *tetA*, *tetE* [36]. A pure colony of resistant isolate with template DNA was added into the reaction mix composed of 1 μL of each primer (except *tet* genes, where was the volume of primers 0.5 μL) and DNAfree PCR water in a total volume of 25 μL. Primers used during each PCR are listed in Table 2. Reaction mixtures were properly vortexed and inserted into the thermocycler (Mastercycler personal Eppendorf, Hamburg, Germany) using the following conditions for amplifying specific sections in β-lactamases: initial denaturation at 94 °C for 20 min.; decrease at 72 °C and HOLD; 30 cycles of denaturation at 94 °C for 40 s., annealing temperature 54 °C for 1 min, 72 °C for 1 min and 30 s, and the final elongation step at 72 °C for 10 min. Conditions used to determine tetracycline genes were: initial denaturation at 94 °C for 20 min.; decrease at 72 °C and HOLD; 35 cycles of denaturation at 94 °C for 40 s., annealing temperature 55 °C for 1 min., 72 °C for 1 min. 30 s, and the final elongation step at 72 °C for 10 min. After initial denaturation, a Master mix (Biotechrabbit, Berlin, Germany) for multiplex PCR was added to the mixture in both cases [34].

PCR products were visualized by gel electrophoresis (1.5% agarose in TAE buffer) set up at 100 V for 1 h and 45 min. Agarose gel was additionally stained by Gel Red (Biotium, Fremont, CA, USA) in TAE solution for 30 min. Positive controls used during the PCR reaction were subjected to sequence analyses to prove the presence of specific resistance genes.

## 3. Results and Discussion

### 3.1. Antibiotic-Resistant Isolates Identification

Samples of raw milk collected from vending machines and samples of smoothies from fresh markets in Slovakia were subjected to monitoring for antibiotic-resistant coliform bacteria. Initial monitoring revealed the number of total coliform bacteria in the smoothie samples ranged from 2.0 ± 0.08 log CFU/mL to 4.2 ± 0.25 log CFU/mL and in the raw milk samples from 2.5 ± 0.04 log CFU/mL to 4.2 ± 0.12 log CFU/mL. *Escherichia coli* was detected in each sample of raw milk and only one sample of smoothie drink. Antibiotic-resistant coliform bacteria were detected in high numbers in both types of smoothies (fruit- or green-based). The most prevalent antibiotic resistance was ampicillin resistance in both types of samples, followed by samples of AO tetracycline resistance and samples of PO gentamicin resistance, respectively [13,14]. Antibiotic-resistant strains were randomly isolated. The collection of 110 antibiotic-resistant coliform bacteria was represented by 30% of milk isolates and 70% of smoothie isolates, where 52% of smoothie isolates came from fruit-based smoothies and 48% from green-based smoothies (Table 3). The majority was identified as *Enterobacter* spp. (45%), *E. coli* (16%) and *Klebsiella* spp. (15%) (Figure 1).

*Enterobacter* was observed in 92% of smoothie samples. The identified isolates were *E. asburiae* (47%), *E. cloacae* (33%), and *E. ludwigii* (20%). *Enterobacter* spp. can be found ubiquitous in the environment [37] and is commonly harbored in plants [38], which are the main ingredients during the preparation of smoothie drinks (spinach, lettuce, etc.). Aside from the ubiquitous occurrence of *Enterobacter* spp., nosocomial infections caused by this bacterium have been described repeatedly over time, and the contribution to spreading resistance is also significant [39,40].

*Escherichia coli* (13%) was predominantly isolated from AO samples, raw milk (83%). Sixty-seven percent of *E. coli* was collected during the summer season due to the conditions which are more favorable for its reproduction [41]. Seasonal changes in milk composition have a huge impact on the microbial load of raw milk [41,42]. The bacterial strain *E. coli* present in milk is a part of the contaminating microbiota [43,44]. It has the potential to be a causative agent for mastitis infections in cows as well as *Staphylococcus aureus* [45]. The occurrence of *E. coli* in raw milk is not only connected to infections such as mastitis but also affects the quality of milk as it degreases and complains of further processing. Moreover, spreading antimicrobial resistance among *Enterobacterales*, with *E. coli*, as one of the agents causing mammary gland infections of dairy cows, highlight the usage of ATBs in such a treatment [46].

All antibiotic-resistant strains of *Klebsiella* spp. Were isolated from food samples of PO-smoothies.

*Raoultella* spp. was identified in 8% of isolates, predominantly detected in samples of raw milk (89%). This bacterium is one of the very rare nosocomial pathogens commonly found in nature. The risk of infection caused by this agent is low, with infections being reported mainly in immunocompromised patients [47,48,49].

Five percent of identified coliform bacteria form bacteria *Hafnia alvei*, which were only detected in samples of raw milk. Four species out of six were isolated from Petri dishes with the addition of gentamicin. The remaining two bacteria were isolated from agar enriched with chloramphenicol. *H. alvei* was formerly known as *Enterobacter hafnia*. Most strains are considered saprophyte [50], but there are also recorded cases of becoming unusual nosocomial pathogens mainly associated with hemolytic uremic syndrome infections [51].

*Serratia* spp. was identified in 4%; *Citrobacter* spp. was recorded in 2%, *Morganella morganii* 2%, *Cronobacter sakazakii* 2%, *Lelliottia amnigena* 1%.

### 3.2. Antibiotic Susceptibility Profile of Isolated Coliform Bacteria

The susceptibility to six different ATB classes was studied using three different concentrations—the first and second reflecting resistance breakpoints according to EUCAST and CLSI. The third was higher than the breakpoints and pointed out on high-level resistance. The growth of all 110 resistant isolates was detected using different concentrations of ampicillin (Table 3). The comparison of parallel analyzes of each isolate did not reveal a significant difference (*p* < 0.005). Eighteen isolates were identified as *E. coli*, predominantly isolated from samples of raw milk. All displayed a high level of ampicillin resistance. The results of this study correspond to the statement, as most coliform bacteria, as a member of the group *Enterobacterales*, report to have intrinsic resistance to ampicillin, i.e., β-lactam ATB (penicillin) [52]. Only isolates of *E. coli* are supposed to be susceptible to ampicillin, as bacterium produces constitutive in a small number of chromosomal β-lactamases AmpC [52,53]. Resistance to ampicillin among *E. coli* in the food chain has been recorded before, and it seems to be spreading intensively [54,55].

A high level of ceftazidime resistance was observed in 35% of resistant coliform isolates, while more than half were represented by *Enterobacter* spp. species. Other ceftazidime-resistant coliforms were *R. planticola* (21%), *E. coli* (18%), *K. oxytoca* (5%), and *S. odorifera* (3%). Ceftazidime resistance was detected in 55% of isolates originating in samples of raw milk in comparison with 31% occurrences detected in isolates of smoothie drinks samples. Ceftazidime is an ATB of third-generation cephalosporins used against many types of gram-negative bacterial infections [56]. *Enterobacter* spp. has intrinsic resistance to β-lactams due to the production of chromosomal AmpC β-lactamases. Exposure to ATB may lead to mutations and selection of strains with permanent β-lactamases hyperproduction (AmpC-overproduction mutants) [57,58]. For example, *Enterobacter* spp.-causing bacteremia had an increasing trend in developing third-generation cephalosporins-resistance in patients who previously received such treatment, compared to patients treated with other ATBs [58].

Chloramphenicol resistance was detected in 31% of all isolates, where most of it (62%) was isolated from smoothie drinks. In comparison, Godziszewska et al. (2018) focused on spreading resistance among coliform bacteria in raw milk, and resistance to chloramphenicol was identified in 78% of all cases [59]. The use of this ATB is limited in the European Union for use in food-producing animals. Chloramphenicol is ATB naturally produced by soil bacteria, which could contribute to higher occurrence of resistance between PO resistant isolates [60].

Resistance to aminoglycoside gentamicin has been recorded in 28% of all resistant isolates. Mostly low-level resistance has been noticed except for two cases of highly resistant strains, specifically *E. ludwigii* and *C. gillenii*. Most gentamicin-resistant isolates were isolated from smoothie drinks (87%). This may be a consequence of the possible practice of using wastewater sludge mixed with compost as fertilizer for plants in Slovakia. In wastewater sludge, gentamicin resistance has been observed in rising volume [61]. According to Štefunková et al. (2020), gentamicin-resistant coliforms were detected in rivers and lakes in Slovakia as well [62], which could contribute to the dissemination of this resistance.

Low-level tetracycline resistance has been observed in 13% of cases, and all isolated strains of *R. planticola* were tetracycline-resistant. More than half of the detected tetracycline resistance belonged to isolates of AO (57%). Fifty-five percent of tetracycline-resistant coliforms were identified in a Polish study performed on raw milk as well [59]. Bacterium *R. planticola* is rarely connected with infection. However, it was observed that it might acquire multiple resistance genes, such as the *bla*_NDM-1_ gene, associated with higher mortality and ineffectiveness of ATB treatment [63].

In the case of detected ciprofloxacin resistance, 7% was recorded as only two isolates were low-resistant, two high-resistant, and five highly resistant. All ciprofloxacin-resistance isolates originate in samples of smoothie drinks. Highly resistant were two strains of *K. oxytoca* and *E. cloacae* and one resistant strain of *C. gillenii*. Low-level resistance to fluoroquinolones has been observed in food previously between *Enterobacterales*. Ciprofloxacin belongs to synthetic ATBs, where resistance is spreading as specific mutations in drug efflux or entry [18].

A high level of meropenem resistance was detected in 5% of resistant coliforms. ATB meropenem is generally marked as last resort ATB, which is commonly used to treat infection caused by multi-resistant strains [18].

MDR has been registered in 36 (33%) isolates, with the dominance of the *Enterobacter* spp. species. Fifty-eight percent of MDR strains have been isolated from samples of smoothie drinks, but with respect to their dominance among isolated strains, it represents 27% of MDR for all PO isolates. On the other hand, 42% of MDR represents isolates from raw milk samples, but regarding the number of isolated strains originated in raw milk, it represents 45% of them.

### 3.3. Antibiotic Resistance Mechanism

**Biofilm:** Biofilm can be described as the formation of a population of mono- or multispecies colonies adhering to a surface [22]. The comparison of parallel analyzes of each isolate did not reveal a significant difference (*p* < 0.005). The results of the biofilm formation detection showed that 73% of resistant coliform bacteria were strong biofilm producers. As Figure 2 shows, most of them was *Enterobacter* spp. Twenty-three percent of isolates were evaluated as very strong producers of biofilm. The isolates that formed very strong biofilm were predominantly *Klebsiella* spp. (36%) and *Enterobacter* spp. (28%). The rest of the very strong producers were *Serratia* spp. (16%) and *E. coli* (16%), plus one isolate of *L. amnigena*. A moderate ability to form biofilm was detected in 4% of cases, specifically 40% in *C. sakazakii* and 60% in isolates *E. coli*. None of the resistant isolates were detected to be low producers of biofilm. The ability to adhere to surfaces is a very important property of infection-causing bacteria in humans [24]. The majority of chronic or recurrent human infections are caused by bacterial biofilms, and *K. pneumoniae* or *E. coli* are a common source of such diseases [22]. In this study, *K. pneumoniae* was very strong biofilm producer. This bacterium is mainly responsible for respiratory tract diseases [64], and the ability to form biofilm makes them even harder to cure. All strains were isolated from smoothie drink samples, which may represent a serious problem due to the preparation of such foodstuffs. Mixing the ingredients together generates aerosols containing microbes, which are easy to transport to the respiratory system of the handler.

**Efflux pump overproduction:** Efflux pump overproduction was clarified in only eight bacteria (7%), and the comparison of the parallel analyzes of each isolate did not reveal a significant difference (*p* < 0.005). The majority of strains belonged to *H. alvei* (6) isolated from milk samples. The rest of the detected efflux pump overproduction was identified in resistant isolates, *E. asburiae* (1) and *S. liquefaciens* (1), both of which were obtained from smoothie samples. The genome of *Hafnia* spp. can acquire resistance genes to multiple ATBs and encode efflux pump genes related to MDR. The efflux pump system is mostly mediated via *farB* and *acrAB-tolC* genes in *Hafnia* spp. [65]. Species of genus *Enterobacter* spp. can develop various resistant mechanisms acquired from the environment or by mutation during medical treatment. Modification of efflux pump system in *Enterobacter* spp. was recorded previously, and resistance mechanism was described predominantly via *acr-AB-tolC* and *ompC* genes. [66]. Genus *Serratia* spp. possesses high intrinsic resistance, which complicates ATB treatment [67]. The active efflux pump system was described mainly in *S. marcescens* through *macAB* efflux system [68].

### 3.4. Important Antibiotic Resistance Genes in Isolated Coliform Bacteria

β-lactam ATBs belong to the most frequent treatment choice as an antibacterial drug for infections in humans, as well as in veterinary medicine [69]. Resistance to β-lactam ATBs is most often mediated through β-lactamases, which inactivate ATBs by hydrolysis of the β-lactam ring. ESBL has arisen as a rapid response to the application of broad-spectrum β-lactam ATBs, which were developed for serious infections caused by gram-negative bacteria. ESBL are derived from narrow-spectrum β-lactamase groups and are predominantly encoded on the mobile genetic element called plasmids [70]. The efflux pumps resistance mechanism plays an important role in MDR, and the transfer could be distributed both vertically and horizontally via plasmids and transposons [71]. The *Enterobacterales* family plays an important role in the transfer of resistance genes between or within bacterial species [72]. According to this fact, antibiotic resistance genes encoding β-lactamases (*bla*_TEM_; *bla*_OXA_; *bla*_SHV_; *bla*_CTX-M_) and efflux pumps (*tetA*, *tetE*) were studied in antibiotic-resistant isolates. The antibiotic resistance profile indicated which of the detected genes should be detected in resistant isolates.

All 110 resistant isolates collected from samples of smoothie drinks and raw milk have recorded high-level resistance to β-lactam ampicillin. Table 4 shows that the resistance gene *bla*_OXA_ was the most prevalent in our isolates. Half of the bacteria with this gene belonged to *E. coli*. The majority were isolated from raw milk (93%), as well as strains of *R. planticola* (29%). *Klebsiella* spp. (21%) caring *bla*_OXA_ gene was gained from smoothie drinks. In 19% of isolates was detected *bla*_TEM_ gene. Except for one strain of *E. coli* isolated from smoothie drinks, all *bla*_TEM_ genes were observed in isolates of AO (*R. planticola* 38%, *E. coli* 29%, *Enterobacter* spp. 19%, *H. alvei* 10%). Fourteen percent of isolates have the *bla*_SHV_ gene. Thirty-three percent create microorganisms of *Enterobacter* spp. and *H. alvei*, 27% *K. pneumoniae,* and 7% *E. coli*. Raw milk has been identified as a source of resistance genes in previous studies as well. In the case of *E. coli* isolated from raw milk of mastitis cows, the predominant *bla* gene was repeatedly marked as a gene *bla*_TEM_ [45,73].

High-level ceftazidime resistance was reported in 35% of cases, and 54% were *Enterobacter* spp. The *bla*_CTX-M_ gene was detected in two cases. Both strains were species of *E. asburiae* with plat origin. Both were isolated from the same fresh bar but were found in different smoothies as one has harbored *bla*_CTX-M-1_ and the other one *bla*_CTX-M-2_. ESBL enzymes are most often encoded by the *bla*_CTX-M_ genes and are responsible for resistance to extended-spectrum cephalosporins, as is ceftazidime [28].

In half (47%) of the isolates, a registered presence of more than one resistance gene was observed (Table 4). All strains of *E. ludwigii* of AO, two *H. alvei*, and one *E. coli* isolate from the samples from raw milk, posed *bla*_TEM_ and *bla*_SHV_ simultaneously. A combination of *bla*_SHV_ and *bla*_OXA_ was reported only in one case of *K. pneumoniae* of PO. Eighty-nine percent (16) of all isolated *E. coli* has posed various resistance genes, and in 31%, a combination of the *bla*_TEM_ and *bla*_OXA_ genes. Regarding *E. coli*, a combination of OXA-type and TEM-type β-lactamases has been observed as the most frequent plasmid-borne enzymes associated with resistance to common medical therapy by amoxicillin–clavulanic acid drugs [45,73].

The genes *tetA* and *tetE* were studied in tetracycline-resistant isolates and strains with overproduction of efflux pumps. These genes were only detected in isolates of *R. planticola* isolated from milk samples (Table 4). Isolates of *R. planticola* were stored in a combination of tetracycline genes *tetA* and *tetE,* but detection of resistance genes proved the presence of β-lactamase genes *bla*_TEM_ and *bla*_OXA_ as well. One of the mechanisms of resistance to tetracyclines is through the energy-depending efflux pumps specific to tetracycline, which are encoded by genes *tetA* and *tetE* [36]. However, detection of the overproduction of efflux pumps did not prove its production between isolates of *R. planticola*. Interestingly, resistance genes may be present in a microorganism, but do not always need to be expressed. Nevertheless, unexpressed genes can further spread to other bacteria [74].

Isolates obtained from samples of AO represent a minority of the collection (30%). However, resistance genes have been predominantly detected in coliform bacteria isolated from raw milk (69%), where 94% of isolates of AO posed resistance genes or a combination of resistance genes (Figure 3). This confirms that livestock can serve as a reservoir for resistance genes and may play a role as one of transmission for antibiotic resistance phenotypes on their way to humans [75].

## 4. Conclusions

The main problem associated with antimicrobial resistance is that medical treatment could fail due to the survival advantage for infectious bacteria. Another effect is the limitation of ATBs used during infections or transmission of antibiotic-resistance to the gastrointestinal microbiome, thereby providing a convenient position to acquire resistance genes by germs. Antibiotic-resistant bacteria or their antimicrobial resistance genes are known to spread from animals or plants to humans via the food chain. Foodstuffs may act as a gene pool for bacteria to gain new antibiotic resistance genes and, therefore, indirectly contribute to the problem of antibiotic resistance and human health. The present stud identified that food commodities such as smoothie drinks or raw milk could harbor antibiotic resistance, which could spread to the commensal gut microbiota and disseminate to the environment in feces. Antibiotic resistance profile of 110 coliform bacteria isolated from both types of samples, mainly identified as *Enterobacter* spp. (45%) and *E. coli* (16%) showed various resistant phenotypes. Thirty-three percent of resistant isolates were MDR; efflux pump overproduction was observed in 7%. Resistant isolates were predominantly evaluated as strong producers of biofilm. Resistance gene *bla*_OXA_ was detected in one-quarter of isolates, especially in *E. coli*, and resistance genes *bla*_TEM_ and *bla*_SHV_ were detected in 19% and 14%, respectively. It should be highlighted that samples of this study were consumed raw, which may contribute to the easier transfer of antimicrobial resistance as the bacteria posing resistant phenotype are not additionally eliminated during preparation. The novelty of the study is the characterization of resistant isolates detected in popular smoothie drinks. Results clearly show that although antibiotic-resistant bacteria in raw ready-to-drink foodstuff do not have to represent a health risk for the consumer, they can contribute to the dissemination of antibiotic resistance in the community and environment. This fact shows the urgency regarding the popularity of smoothies and raw milk in Slovakia consumed for the purpose of a healthier life.

## Figures and Tables

**Figure 1 foods-11-01324-f001:**
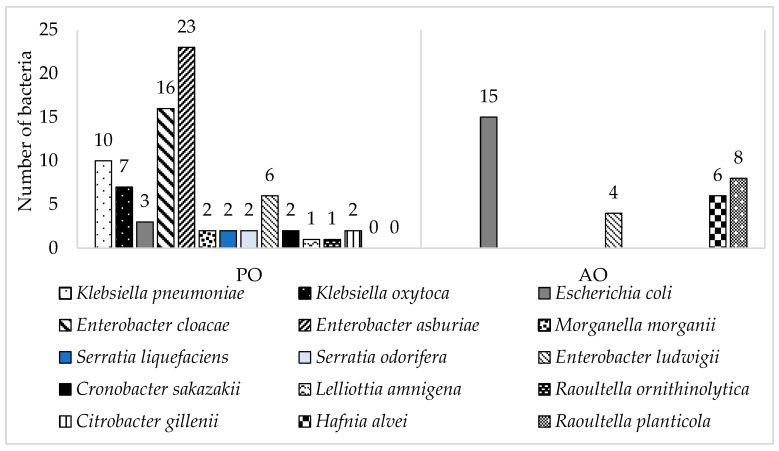
Number of identified resistant coliform bacteria isolated from samples of smoothie drinks (PO) and raw milk (AO).

**Figure 2 foods-11-01324-f002:**
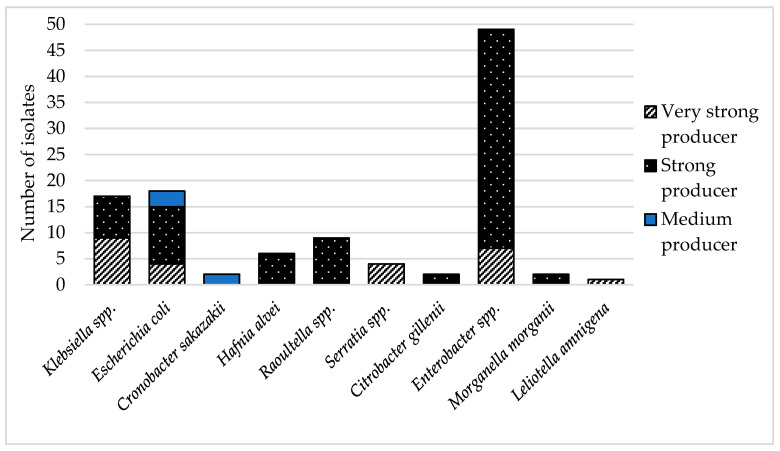
Types of biofilm production of antibiotic-resistant isolates.

**Figure 3 foods-11-01324-f003:**
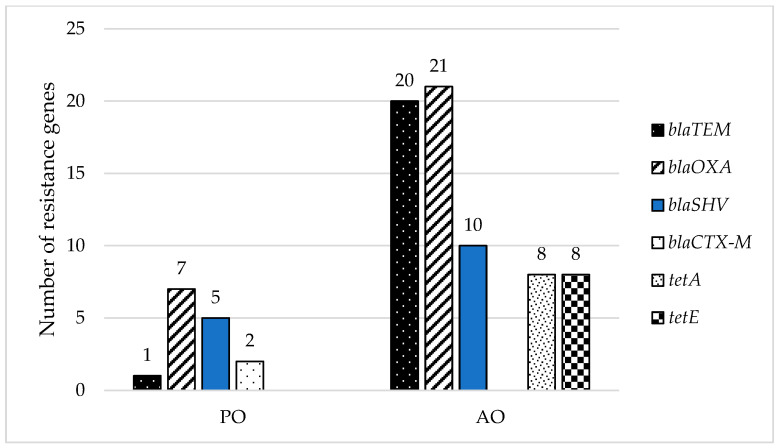
Distribution of detected resistance genes in different types of samples (smoothie drinks-PO and raw milk-AO).

**Table 1 foods-11-01324-t001:** Resistant breakpoints of ATBs used in susceptibility testing.

ATB Class	Antimicrobial	EUCAST (mg/L) >	CLSI (mg/L) ≥	Higher Than CLSI (mg/L)
Penicillins	Ampicillin	8	32	50
Cephalosporins	Ceftazidime	4	16	32
Carbapenems	Meropenem	8	4	12
Fluoroquinolones	Ciprofloxacin	0,5	1	2
Aminoglycosides	Gentamicin	2	16	20
Amphenicols	Chloramphenicol	8	32	50
Tetracyclines	Tetracycline	-	16	32

EUCAST-The European Committee on Antimicrobial Susceptibility Testing; CLSI-Clinical Laboratory Standards Institute.

**Table 2 foods-11-01324-t002:** Resistant genes detected during PCR in resistant coliform isolates.

ATB Class	Gene	Primer	DNA Sequence 5′→3′	AS (bp)	AT (°C)
β-Lactams	*bla*_TEM_ *	fwd	CATTTCCGTGTCGCCCTTATTC	800	54
		rev	CGTTCATCCATAGTTGCCTGAC		
	*bla*_SHV_ *	fwd	AGCCGCTTGAGCAAATTAAAC	713	54
		rev	ATCCCGCAGATAAATCACCAC		
	*bla*_OXA_ *	fwd	GGCACCAGATTCAACTTTCAAG	564	54
		rev	GACCCCAAGTTTCCTGTAAGTG		
	*bla*_CTX-M_ group 1 *	fwd	TTAGGAARTGTGCCGCTGYA	688	54
		rev	CGATATCGTTGGTGGTRCCAT		
Tetracyclines	*tetA* **	fwd	GCTACATCCTGCTTGCCTTC	210	55
		rev	CATAGATCGCCGTGAAGAGG		
	*tetE* **	fwd	AAACCACATCCTCCATACGC	278	55
		rev	AAATAGGCCACAACCGTCAG		

AS—amplicon size; AT—annealing temperature. * Favier et al., 2018 [35]. ** Ng et al., 2001 [36].

**Table 3 foods-11-01324-t003:** Antibiotic susceptibility profile of coliform bacteria isolated from raw milk and smoothie drinks.

Resistant Isolate	Sample	AMP	CIP	GEN	CHF	TET	CEF	MER	MDR	Resistant Isolate	Sample	AMP	CIP	GEN	CHF	TET	CEF	MER	MDR
*E. coli*	PO-GS	R	S	R1	S	S	S	S	-	*L. amnigena*	PO-FS	R	S	S	S	R1	S	S	-
*E. cloacae*	PO-GS	R	S	R1	S	S	S	S	-	*S. liquefaciens*	PO-FS	R	S	S	S	S	S	S	-
*K. pneumoniae*	PO-GS	R	S	R1	S	S	S	S	-	*E. cloacae*	PO-FS	R	R	R1	R1	S	S	S	+
*K. pneumoniae*	PO-GS	R	S	R1	S	S	S	S	-	*E. cloacae*	PO-FS	R	S	S	R1	S	S	S	-
*K. pneumoniae*	PO-GS	R	S	R1	S	S	S	S	-	*E. cloacae*	PO-GS	R	S	R1	R1	S	R	S	+
*K. pneumoniae*	PO-GS	R	S	R1	S	S	S	S	-	*C. gillenii*	PO-GS	R	S	R1	S	S	S	S	-
*K. pneumoniae*	PO-GS	R	S	S	S	S	S	S	-	*K. oxytoca*	PO-GS	R	S	R1	S	S	S	R	+
*K. pneumoniae*	PO-GS	R	S	S	R	S	S	S	-	*R. ornithinolytica*	PO-GS	R	S	S	S	S	S	S	-
*K. pneumoniae*	PO-GS	R	S	S	S	S	S	S	-	*E. ludwigii*	PO-GS	R	S	S	R1	S	R	S	+
*E. coli*	PO-GS	R	S	S	R	R1	S	S	+	*E. cloacae*	PO-GS	R	S	S	R1	S	S	S	-
*E. coli*	PO-GS	R	S	S	R	R1	S	S	+	*E. cloacae*	PO-GS	R	R	R1	R1	S	S	S	+
*K. pneumoniae*	PO-GS	R	S	S	S	S	S	S	-	*C. gillenii*	PO-GS	R	R	R	R	S	S	S	+
*K. pneumoniae*	PO-GS	R	S	S	S	S	S	S	-	*E. cloacae*	PO-GS	R	S	S	S	S	S	S	-
*E. asburiae*	PO-GS	R	S	S	S	S	S	S	-	*E. cloacae*	PO-GS	R	S	S	S	S	S	S	-
*E. cloacae*	PO-GS	R	S	S	S	S	S	S	-	*E. cloacae*	PO-FS	R	S	S	R1	S	S	S	-
*K. pneumoniae*	PO-GS	R	S	S	S	S	S	S	-	*E. cloacae*	PO-FS	R	S	S	R1	S	S	S	-
*M. morganii*	PO-GS	R	S	S	S	S	S	S	-	*E. asburiae*	PO-FS	R	S	S	S	S	S	S	-
*M. morganii*	PO-GS	R	S	S	R1	R1	S	S	+	*E. asburiae*	PO-FS	R	S	S	R1	S	S	S	-
*K. oxytoca*	PO-FS	R	R	R1	R1	R1	R	R	+	*S. liquefaciens*	PO-FS	R	S	S	S	S	S	S	-
*K. oxytoca*	PO-FS	R	S	S	S	S	S	S	-	*E. asburiae*	PO-FS	R	S	S	S	S	S	S	-
*K. oxytoca*	PO-FS	R	S	S	R1	S	S	S	-	*E. ludwigii*	PO-FS	R	R1	S	R1	S	R	S	+
*K. oxytoca*	PO-FS	R	R	R1	R	S	R	R	+	*E. ludwigii*	PO-FS	R	S	S	R1	S	S	S	-
*E. asburiae*	PO-FS	R	S	S	S	S	R	S	-	*E. coli*	AO	R	S	S	S	S	S	S	-
*E. asburiae*	PO-FS	R	S	S	S	S	R	S	-	*E. coli*	AO	R	S	S	S	S	S	S	-
*E. asburiae*	PO-FS	R	S	S	S	S	R	S	-	*E. coli*	AO	R	S	S	S	S	S	S	-
*E. asburiae*	PO-GS	R	S	R1	S	S	R	S	+	*E. coli*	AO	R	S	S	S	S	S	S	-
*E. asburiae*	PO-GS	R	S	R1	S	S	R	S	+	*H. alvei*	AO	R	S	S	R1	S	S	S	-
*E. asburiae*	PO-GS	R	S	S	S	S	R	S	-	*H. alvei*	AO	R	S	S	R1	S	S	S	-
*E. asburiae*	PO-GS	R	S	S	S	S	R	S	-	*H. alvei*	AO	R	S	S	R1	S	S	S	-
*E. asburiae*	PO-GS	R	R2	R1	R1	S	R	R	+	*H. alvei*	AO	R	S	S	R1	S	S	S	-
*E. asburiae*	PO-GS	R	S	S	S	S	S	S	-	*H. alvei*	AO	R	S	S	R1	S	S	S	-
*E. asburiae*	PO-GS	R	S	R1	R1	S	R	S	+	*H. alvei*	AO	R	S	S	R1	S	S	S	-
*E. asburiae*	PO-GS	R	S	S	S	S	R	S	-	*E. ludwigii*	AO	R	S	R1	S	S	S	S	-
*E. asburiae*	PO-GS	R	S	S	S	S	S	S	-	*E. ludwigii*	AO	R	S	R1	S	S	S	S	-
*E. asburiae*	PO-FS	R	S	S	S	S	S	S	-	*E. ludwigii*	AO	R	S	R1	S	S	S	S	-
*E. asburiae*	PO-FS	R	S	S	S	S	R	S	-	*E. coli*	AO	R	S	R1	S	S	S	S	-
*E. asburiae*	PO-FS	R	S	S	S	S	R	S	-	*E. ludwigii*	AO	R	S	S	S	S	S	S	-
*E. asburiae*	PO-FS	R	S	S	S	S	R	S	-	*E. coli*	AO	R	S	S	S	S	R1	S	-
*E. asburiae*	PO-FS	R	S	S	S	S	R	R	+	*E. coli*	AO	R	S	S	S	S	R1	S	-
*E. asburiae*	PO-FS	R	S	S	S	S	R	S	-	*E. coli*	AO	R	S	S	S	S	R1	S	-
*E. asburiae*	PO-FS	R	S	R1	S	R1	R	S	+	*R. planticola*	AO	R	S	S	S	R2	R2	S	+
*K. oxytoca*	PO-FS	R	S	S	S	S	S	S	-	*R. planticola*	AO	R	S	S	S	R2	R2	S	+
*K. oxytoca*	PO-FS	R	S	S	S	S	S	S	-	*R. planticola*	AO	R	S	S	S	R2	R2	S	+
*E. cloacae*	PO-FS	R	S	R1	S	S	R	S	+	*R. planticola*	AO	R	S	S	S	R2	R2	S	+
*E. cloacae*	PO-FS	R	R1	R1	S	S	S	R	+	*R. planticola*	AO	R	S	S	S	R2	R2	S	+
*S. odorifera*	PO-FS	R	S	R1	S	S	R	S	+	*R. planticola*	AO	R	S	S	S	R1	R1	S	+
*S. odorifera*	PO-FS	R	S	R1	S	S	S	S	-	*R. planticola*	AO	R	S	S	S	R1	R1	S	+
*E. cloacae*	PO-FS	R	S	S	S	S	R	S	-	*R. planticola*	AO	R	S	S	S	R1	R1	S	+
*E. cloacae*	PO-FS	R	S	S	S	S	S	S	-	*E. coli*	AO	R	S	S	R1	S	R2	S	+
*E. ludwigii*	PO-FS	R	S	S	S	S	S	S	-	*E. coli*	AO	R	S	S	R1	S	R2	S	+
*E. ludwigii*	PO-FS	R	S	S	S	S	S	S	-	*E. coli*	AO	R	S	S	R1	S	R2	S	+
*E. cloacae*	PO-FS	R	S	R1	S	S	S	S	-	*E. coli*	AO	R	S	S	R1	S	R2	S	+
*E. ludwigii*	PO-FS	R	S	R	S	S	S	S	-	*E. coli*	AO	R	S	S	R1	S	R2	S	+
*C. sakazakii*	PO-FS	R	S	R1	S	S	S	S	-	*E. coli*	AO	R	S	S	R1	S	R2	S	+
*C. sakazakii*	PO-FS	R	S	R1	S	S	S	S	-	*E. coli*	AO	R	S	S	R1	S	R2	S	+

PO—plant origin; GS—green-based smoothie; FS—fruit-based smoothie; AO—animal origin. AMP—ampicillin; CIP—ciprofloxacin; GEN—gentamicin; CHF—chloramphenicol; TET—tetracycline; CEF—ceftazidime; MER—meropenem; MDR—multidrug-resistant. S—susceptible; R1—concentration of ATB according to European Committee on Antimicrobial Susceptibility Testing; R2—concentration of ATB according to Clinical and Laboratory Standards Institute; R—concentration of ATB showing high level resistance.

**Table 4 foods-11-01324-t004:** Specific resistance genes in isolates of food chain detected by multiplex PCR.

Isolates Positive for Resistance Genes	Sample	β-Lactamase Genes						ESBL		*Tet* Genes	
		*bla* _TEM_	*bla* _SHV_	*bla* _OXA_	*bla*_TEM_ + *bla*_SHV_	*bla*_TEM_ + *bla*_OXA_	*bla*_SHV_ + *bla*_OXA_	*bla* _CTX-M-1_	*bla* _CTX-M-2_	*tetA*	*tetE*
*E. coli* (*n* = 2)	PO	1		1							
*E. cloacae* (*n* = 1)	PO		1								
*K. pneumoniae* (*n* = 5)	PO		4	2			1				
*K. oxytoca* (*n* = 4)	PO			4							
*E. asburiae* (*n* = 2)	PO							1	1		
*E. coli* (*n* = 14)	AO	6	1	13	1	5					
*H. alvei* (*n* = 5)	AO	2	5		2						
*E. ludwigii* (*n* = 4)	AO	4	4		4						
*R. planticola* (*n* = 8)	AO	8		8		8				8	8

PO—plant origin; AO—animal origin.

## Data Availability

Data of the current study are available from the corresponding author. The data are not publicly available due to privacy.

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
