# Peer review of "Characterization of Antibiotic Resistant Coliform Bacteria and Resistance Genes Isolated from Samples of Smoothie Drinks and Raw Milk"

_foods, 2022, doi:10.3390/foods11091324_

Round 1
Reviewer 1 Report
Manuscript ID: foods-1678623
Characterization of antibiotic resistant coliform bacteria and resistance genes isolated from samples of smoothie drinks and raw milk. Monika Krahulcová, Lucia Bírošová, Klára Cverenkárová, Petra Olejníková, Barbora Micajová.
The manuscript provides valuable information and may be published in "Foods". The materials and methods section is quite adequately presented. The results are presented in a clear and comprehensible way. I have only minor remarks.
Abbreviations used for the first time in the text require an explanation, eg HUS, CLSI, EUCAST.
A graphical abstract could support further the study.
Please add also a complete list of abbreviations.
Based on the above, I suggest a minor revision.
Author Response
Thank you very much for your suggestions.
We have completed the minor revision such as abbreviations explanation and graphical abstract.
Reviewer 2 Report
The manuscript is interesting and well written. However, there are some minor comments to be revised to clarify and improve some points in the manuscript. The methods must be better described in the material and methods section. There are several typos in the manuscript, the authors must revise it.
Figure 2 is unnecessary, I recommend deleting it. Figures 1, 3 and 4 could be improved in quality (black color in lines and text).
Other than that, the paper is of good quality.
Author Response
Thank you very much for your suggection.
We have deleted Fig. 2. We have changed the color of the rest of figures in black to improve the quality (in lines and in text).
If you want to see improved manuscript, please see the attachment.

Reviewer 3 Report
Manuscript entitled Characterization of antibiotic resistant coliform bacteria and resistance genes isolated from samples of smoothie drinks and raw milk describes an interesting study on the evaluation of smoothie drinks and raw milk for coliform possessing antibiotic resistance genes. In addition, other abilities of coliform bacteria such as biofilm formation and efflux pumps overproduction were examined. Below please find my review along with tips and suggestions for this manuscript:
- L 69 - 70 In my opinion the statement that coli is one of the most pathogenic bacteria in the environment is highly disputable. Please better substantiate this fact with recent publications or use different phrase.
- 170-172; Please point that not only E. coli are one of the main factor causing mammary gland but also for example Staphylococcus aureus and other pathogens.
- The introduction section says nothing about the relation of coliform bacteria to biofilm formation and efflux pumps overproduction. In my opinion this is worth mentioning in the introduction to let the reader knows for what these tests were done. In fact it is described in L257 - L263 and L278 - L283 but I think it would be better for the understanding of the manuscript if it will be mentioned before result section.
- Please describe in more details the origin of the smoothie juices, whether they were processed in some way or freshly made in fresh bars and intended to be consumed in a short time. Were they made on site or transported from outside suppliers? Please describe in all details the specific characterization of used beverages, including composition, obtaining method, producer name, shelf life, method of extending the shelf life, because this aspects are crucial for microbial quality.
- Giving only literature sources in Materials and methods section is definitely not enough. Used methods are poorly or even not described. If other scientist would like to replicate describe research, she/he should read at least 6-8 other publications. It would be much more smooth if authors could describe used methods in presented manuscript.
- Again: subsection 2.2 & 2.3 - Identification of antibiotic-resistant strains - The method for identifying microorganisms was very poorly described in this manuscript. It is true that a reference has been provided where the methodology is described. However, I think that omitting the description of the stage of culturing microorganisms on selective media is a gross oversight on the part of the authors. I would suggest supplementing the manuscript with this part of the methodology, as well as a more detailed description of the principle of the method for identification of microorganisms using MALDI-TOF.
- Results presented in figure 1 would be much more easy to read and interpretation if authors could present it in a table. Please change.
- Again: subsection 2.6 - How the genetic material was isolated for further PCR procedures? In my opinion, providing a reference with the methodology used in other studies is insufficient. Please describe the genes detection methods in details. What primers were used for the PCR reaction? What were the conditions of the PCR reaction. A table with the sequences of the primers used would be welcome (in supplementary materials, for example). How were the positive controls sequenced? Are these sequences available somewhere? This subsection must be improved.
- Table 1 - Please explain the abbreviations EUCAST and CLSI
- Subsection 3.3 - In Materials and Methods subsection the order of description of analyses is as follows: first biofilm production and then efflux pumps overproduction. For the continuity, I think it would be better to keep the same order also in Results and Discussion section.
- The last sentence in Conclusion section is too general. Please explain in more details what the results of this research contribute to science and practice. How this results can be used in further research/application? Perhaps the authors have an idea how to solve the problem of spreading antibiotic resistance genes in raw products without having to give up eating them? It should be clarify.
- The References section is very poorly prepared. In the references section, out of 78 references, at least 34 have errors such as missing journal name, lack of italics in species names, full journal names although they should be abbreviated. A reference list prepared in this manner cannot be accepted and is a huge disadvantage of this manuscript. I recommend a major review of the reference index and improvement according to the guide for authors. THERE ARE SO MANY MISTAKES/ERROR TYPES IN LITERATURE THAT I WAS REALLY DISAPPOINTED DURING READING.
- Presented results are lack of any statistical analysis, this kind of scientific results should be statistical supported.
- In my opinion, the aim of the research is not correctly formulated. The authors state that the aim of the study was to characterize the resistance profile of isolates to selected antibiotics. However, while reading the article, I got the impression that the aim was initially to isolate and characterize the microflora of raw milk and smoothie samples and then identification of specific genes.
- The authors describe raw milk and smoothie as sources of antibiotic-resistant bacteria. But after all, the identified isolates do not confirm the pathogenicity characteristics, so how to explain the validity of using antibiotics against these isolates?
Author Response
Thank you very much for your suggestion.
Please, see the attachment.

Round 2
Reviewer 3 Report
Dear Authors,
thank you for your corrections. I think that current version of your manuscript is well now, however the last sentence "This fact shows the urgency with regard to the popularity of smoothies and raw milk in Slovakia consumed for the purpose of a healthier life" is in my opinion out of sense and manuscript goal.